# Transcriptomic Characterization of Key Factors and Signaling Pathways for the Regeneration of Partially Hepatectomized Liver in Zebrafish

**DOI:** 10.3390/ijms25137212

**Published:** 2024-06-29

**Authors:** Guili Song, Guohui Feng, Qing Li, Jinrong Peng, Wei Ge, Yong Long, Zongbin Cui

**Affiliations:** 1Key Laboratory of Breeding Biotechnology and Sustainable Aquaculture, Institute of Hydrobiology, Chinese Academy of Sciences, Wuhan 430072, China; 2College of Animal Sciences, Zhejiang University, Hangzhou 310058, China; 3Department of Biomedical Sciences and Centre of Reproduction, Development and Aging (CRDA), Faculty of Health Sciences, University of Macau, Taipa, Macau SAR 999078, China; 4Guangdong Provincial Key Laboratory of Microbial Culture Collection and Application, State Key Laboratory of Applied Microbiology Southern China, Institute of Microbiology, Guangdong Academy of Sciences, Guangzhou 510070, China

**Keywords:** zebrafish, partial hepatectomy, liver regeneration, gene expression, RNA-seq

## Abstract

Liver regeneration induced by partial hepatectomy (PHx) has attracted intensive research interests due to the great significance for liver resection and transplantation. The zebrafish (*Danio rerio*) is an excellent model to study liver regeneration. In the fish subjected to PHx (the tip of the ventral lobe was resected), the lost liver mass could be fully regenerated in seven days. However, the regulatory mechanisms underlying the liver regeneration remain largely unknown. In this study, gene expression profiles during the regeneration of PHx-treated liver were explored by RNA sequencing (RNA-seq). The genes responsive to the injury of PHx treatment were identified and classified into different clusters based on the expression profiles. Representative gene ontology (GO) enrichments for the early responsive genes included hormone activity, ribosome biogenesis and rRNA processing, etc., while the late responsive genes were enriched in biological processes such as glutathione metabolic process, antioxidant activity and cellular detoxification. The Kyoto Encyclopedia of Genes and Genomes (KEGG) pathway enrichments were also identified for the differentially expressed genes (DEGs) between the time-series samples and the sham controls. The proteasome was overrepresented by the up-regulated genes at all of the sampling time points. Inhibiting proteasome activity by the application of MG132 to the fish enhanced the expression of Pcna (proliferating cell nuclear antigen), an indicator of hepatocyte proliferation after PHx. Our data provide novel insights into the molecular mechanisms underlying the regeneration of PHx-treated liver.

## 1. Introduction

The liver is pivotally positioned in the regulation of whole-body metabolism, synthesis of blood protein and clotting factors, storage of glycogen, secretion of bile and homeostasis, as well as in compound detoxification. Consequently, the liver has evolved a phenomenal capacity for compensatory growth and/or regeneration to respond to chemical, traumatic or infectious injuries, and this phenomenon allows liver to recover the lost mass and functions without imperiling the viability of the entire organism [1,2]. Liver functions must be rapidly restored after liver transplantation or resection [3]. Therefore, it is imperative to improve the understanding of processes that constitute normal liver regeneration and how to safely stimulate them.

It has been documented that the type of liver proliferative response is dependent on the degree of liver mass loss in rodents, with a liver mass loss of 70% leading to a strong regenerative response throughout the entire liver remnant and small local liver injuries resulting in a local response [4]. Liver regeneration in mammals is usually investigated by a surgical procedure to remove three of the five liver lobes, which represents two-thirds of the liver mass and is known as two-thirds PHx. The original liver mass and functions are recovered within the following 5–7 days by a compensatory growth of the remaining liver lobes until the original organ weight is reached, but the ablated liver lobes never grow back [5,6]. These complex processes are spatiotemporally and precisely regulated by coordinated interactions among the hepatocytes [7], liver sinusoidal endothelial cells (LSECs) [8,9], Kupffer cells [10] and hepatic stellate cells (HSCs) [11]. A large number of previous studies have revealed the involvement of factors and molecular events such as cytokines and growth factors, matrix remodeling, metabolic signals and functional compensation during liver regeneration [3,12,13]. The mechanisms that trigger the initiation of liver regeneration after PHx and how the liver integrates different signaling pathways and metabolic stresses to control cell proliferation, and stops once the liver reaches its original mass to avoid an oscillatory cycle of overgrowth and apoptosis, remain largely unknown.

Zebrafish are an excellent model to study liver regeneration [14,15]. There are many approaches to induce liver regeneration in zebrafish, such as moderate PHx, acute alcohol or CCl4 exposure. In these cases, the remnant hepatocytes proliferate to recover the lost liver mass [16,17,18]. The liver of adult zebrafish consists of three lobes, including one ventral lobe and two dorsal lobes [19]. We have previously investigated the molecular events during the compensatory growth of zebrafish liver after one-third PHx, namely removing the whole ventral lobe leading to a loss of one-third of the liver weight [20], which were strikingly similar to those of rodents and humans after two-thirds PHx [21]. Analogously, the type of regenerative response in the zebrafish liver is also dependent on the size of the injury to the liver; 7 days after local injury caused by removing the caudal portion of the ventral lobe, the liver is indistinguishable from the sham-operated animals with the caudal portion of the ventral lobe completely regenerated [17].

During the past few years, studies in zebrafish had uncovered several key genes that regulate liver regeneration. The *uhrf1^+/−^* adults did not fully regrow after liver resection, probably due to its role in liver cell proliferation [17]. The *top2a* gene, involved in chromosome decatenation, also has been reported to play important roles in regulating hepatocyte proliferation and liver regeneration [22]. Recently, the Calpain 3b (*Capn3b*), a Ca^2+^ −dependent cysteine proteinase was found to regulate embryonic liver growth and regeneration, based on the finding that the liver regeneration was delayed with accumulating G2/M transition inhibitors Chk1 (checkpoint kinase 1) and Wee1 (WEE1 G2 checkpoint kinase) [23]. Although the previous studies revealed the functions of a few key genes, gene expression dynamics during liver regeneration induced by PHx remains largely unknown.

In this study, we characterized the dynamics of gene expression in the PHx-treated liver of zebrafish. Samples were collected at different time points after the PHx surgery and subjected to RNA-seq. The results revealed that large numbers of genes were differentially expressed during liver regeneration. The GO and KEGG pathway enrichments for the DEGs were identified to shed light on the biological processes and pathways associated with liver regeneration in zebrafish. The early responsive genes were enriched in the biological processes such as hormone activity, ribosome biogenesis and rRNA processing, while the late responsive genes were enriched in physiological functions such as glutathione metabolism, antioxidation and cellular detoxification. The pathway proteasome was found to be overrepresented by the up-regulated genes all of the sampling time points. A negative role of proteasome in regulating liver regeneration was revealed by the application of the specific inhibitor MG132. These data furthered our understanding of the molecular mechanisms underlying liver regeneration. 

## 2. Results

### 2.1. Liver Regeneration after Partial Hepatectomy

The liver regeneration of the *Tg(fabp10a:dsRed; ela3l:EGFP)* transgenic zebrafish after the PHx surgery was explored by fluorescence microscopy. The newly grown liver tissue had brighter red fluorescence than the remnant tissue (Figure 1). At 7 d after the PHx surgery, the newly grown portion reached one-third of the total length of the ventral liver lobe (the horizontal line in Figure 1), suggesting the accomplishment of liver regeneration.

### 2.2. Characterization of Gene Expression during Liver Regeneration

High-throughput sequencing generated 8.23–22.23 million (M) read pairs for the samples. Reads number and mapping rates for the constructed sequencing libraries are displayed in Appendix A. After filtering with the threshold described in the Materials and Methods section, 12,809 genes were found to be expressed in the liver samples.

According to the distribution curves for average gene transcriptional abundance (Figure 2A), the ratio of low-abundance genes gradually decreased during liver regeneration. For example, about 25% of the expressed genes demonstrated a log10(TPM+0.1) value smaller than zero in the sham samples, while the number for the 7 d samples after the PHx surgery was less than 5% (indicated by the dashed line in Figure 2A). Results of PCA (principal component analysis) for the gene abundance data revealed the overall trend of changes in gene expression in the regenerating liver after the PHx surgery (Figure 2B). The trajectory of the samples in terms of the PC1 and PC2 coordinates consisted of a loop. Namely, the samples collected shortly after the PHx surgery (6 h, 1 d and 3 d) were far away from the sham samples, indicating large-scale of changes in gene expression; however, at the late stages of liver regeneration, coordinates of the PHx samples (5 d and 7 d) gradually approached those of the sham samples (Figure 2B), indicating the recovery of gene expression and near accomplishment of the regeneration process.

The DEGs between the samples collected at different time points and the sham samples were identified. The largest number of DEGs was found at 6 h (3,872) after the PHx surgery, followed by the number at 1 d (3,806), indicating acute responses to the injury; the number of DEGs began to decrease at 3 d (2,654), and had a marked reduction at 7 d (881, Figure 2C), suggesting the completeness of the regeneration process. Furthermore, at each time point, similar numbers of up- and down-regulated genes were obtained (Figure 2C). 

The expressions of 15 genes including *anp32b* (acidic (leucine-rich) nuclear phosphoprotein 32 family, member B), *apoeb* (apolipoprotein Eb) and *calm3b* (calmodulin 3b (phosphorylase kinase, delta)), etc. were analyzed by qPCR to validate the RNA-seq results. The results of qPCR assay validated the up- and down-regulation of all of the genes detected by RNA-seq (Appendix A). A significant correlation (*p* = 1.1394 × 10^−42^) between the RNA-seq and qPCR data (log2Foldchange in gene expression compared to the sham samples) was revealed by linear correlation analysis (Figure 2D). 

The up- and down-regulated genes at 6 h, 1 d, 3 d and 5 d are shown in the volcano plots in Figure 2E. The most significantly (top three) up- and down-regulated genes are indicated with the gene symbols. For example, at 6 h after injury, *odc1* (ornithine decarboxylase 1), *cog2* (component of oligomeric golgi complex 2) and *ptp4a2b* (protein tyrosine phosphatase 4A2b) are the most significant up-regulated genes while *akt1s1* (AKT1 substrate 1 (proline-rich)), *ugl* (ureidoglycolate lyase) and *xirp2a* (xin actin binding repeat containing 2a) are the most significant down-regulated genes.

### 2.3. GO Enrichments of the DEG Clusters

The DEGs were classified into 10 groups with K-means clustering. According to the expression patterns, most genes demonstrated transient up- or down-regulation at certain stages after the injury (Figure 3A). For example, cluster 7 was transiently up-regulated at 6 h and cluster 8 was shortly up-regulated at 1 d. There were also down-regulated gene clusters. Cluster 9 was transiently down-regulated from 6 h to 1 d, and cluster 2 was consistently down-regulated from 6 h to 7 d after the injury (Figure 3A). To demonstrate variation patterns of gene expression, the representative gene clusters were subjected to weighted gene co-expression network analysis (WGCNA); most of the genes in each cluster were contained in the major module. The module eigengene values illustrate overall expression profiles of the genes in clusters 3, 5, 6 and 8 (Appendix A). 

The DEG clusters were subjected to GO enrichment analysis and GO enrichments for the DEG clusters are detailed in Appendix A. For the clusters consisting of early and transiently up-regulated genes at 6 h after the PHx surgery, cluster 1 was enriched in transmembrane transport and hormone activity, both cluster 5 and cluster 7 were enriched in ribosome biogenesis and rRNA processing and cluster 5 was also enriched in RNA modification, RNA transport and RNA splicing (Figure 3B). For the gene clusters down-regulated during liver regeneration, cluster 2 was enriched in response to estrogen and cytoplasmic translation, cluster 9 was enriched in cellular amino acid catabolic process and organic acid catabolic process and cluster 4 was minimally enriched in DNA-templated histone modification and transcription (Figure 3B). Together, taking into account the expression patterns and corresponding GO enrichments for the DEG clusters at the same time provides further insights into the molecular mechanisms underlying liver regeneration after the PHx surgery. 

### 2.4. KEGG Pathway Enrichments for the DEGs during Liver Regeneration

KEGG pathway enrichments of the up- and down-regulated genes under each condition are listed in Appendix A. For the pathway enrichments of the up-regulated genes (Figure 4A), salmonella infection and the proteasome were identified throughout the experiment and the phagosome was also widely enriched for the up-regulated genes from 1 d to 7 d, suggesting that high activities of these pathways are continuously needed during liver regeneration after injury. Ribosome biogenesis in eukaryotes and protein processing in the endoplasmic reticulum were only enriched among the genes up-regulated at the early stages (6 h and 1 d) after the PHx surgery. Pathways including DNA replication, mismatch repair and tight junctions were overrepresented by the genes up-regulated in the middle stages (1 d to 5 d) of liver regeneration.

Many pathways associated with metabolism and xenobiotics detoxification, such as the metabolism of xenobiotics by cytochrome P450, lysine degradation, pantothenate and CoA biosynthesis and histidine metabolism, are highly enriched among the genes down-regulated at the early stages (6 h to 1 d) after the PHx surgery (Figure 4A), indicating the suppression of liver functions by the injury. The enrichment of pathways such as glycerolipid metabolism, the PPAR (peroxisome proliferator-activated receptor) signaling pathway, glycolysis/gluconeogenesis and ribosomes were among the down-regulated genes from 6 h to 5 d (Figure 4A), indicating the late recovery of these physiological functions after receiving the injury. Except for glycerolipid metabolism, none of the above-mentioned pathways are enriched among the down-regulated genes at 7 d (Figure 4A), suggesting the complete rebuilding of liver functions at this stage.

The genes associated with the representative pathways are shown in Figure 4B. The genes associated with DNA replication, such as *pcna* (proliferating cell nuclear antigen), were up-regulated from 1 d to 5 d after PHx. The proteasome subunits such as *psma1* (proteasome 20S subunit alpha 1) and *psmd2* (proteasome 26S subunit ubiquitin receptor, non-ATPase 2) were highly up-regulated at 1 d and 3 d. Overall, expression patterns of the representative genes reflect activities of the corresponding pathways.

### 2.5. Effects of Modulating Proteasome Activity on Liver Regeneration

The proteasome was overrepresented by the up-regulated genes throughout the experiment (Figure 4A), suggesting crucial functions for the proteasome in liver regeneration. To investigate the roles of the proteasome, we first analyzed protein levels of Psmd2 (a subunit of the 19S regulatory complex of proteasomes) and Pcna. Consistent with the transcriptomic data, both Psmd2 and Pcna were induced by PHx. The expression of Pcna was continuously up-regulated from 6h to 3d and maintained at a high level from 3d to 7d; the highest level of Psmd2 was found at 5d after PHx (Figure 5A). By using Pcna as a marker for cell proliferation and liver regeneration, we characterized the effects of inhibiting proteasome activity through injecting the fish with MG132. The results indicate that the application of MG132 had no impact on the level of Psmd2, but enhanced the level of Pcna at 3 d, 5 d and 7 d after PHx (Figure 5B). Together, both transcript and protein levels of Psmd2 and Pcna were up-regulated after the PHx surgery and inhibiting the proteasome activity with MG132 enhanced the expression of Pcna.

## 3. Discussion

Liver regeneration after local injury occurs in zebrafish and is similar to that in mammals; the initiation of this process is closely associated with the activation and proliferation of hepatocytes [21]. However, the origins of the initial signals for the activation of the liver remnant and how multiple regulators, signaling pathways and cell types are precisely integrated to remodel the injured liver remain largely unknown. In this study, we characterized the dynamics of transcriptional gene expression during the liver regeneration in zebrafish using the PHx model, in which one-third of the ventral liver lobe was resected and the remaining ventral lobe of the liver was subjected to RNA-seq assays. The gene expression of the samples collected at different time points (6 h, 1 d, 3 d, 5 d and 7 d) after the PHx surgery was compared with the sham controls to identify the DEGs and the genes underwent differential splicing.

The samples collected at different time points during liver regeneration demonstrated a circular trajectory in terms of the first two principal components, reflecting the progress from the initial responses to the rebuilding of the physiological organ functions. Large numbers of both up- and down-regulated genes were identified at 6 h after the PHx surgery, indicating acute responses of the cells in the remaining liver tissue to the injury. Looking at the most significantly up-regulated genes can provide information for the molecular events acutely activated by the injury. Among the top genes up-regulated at 6 h after PHx, *odc1* encodes the rate-limiting enzyme catalyzing ornithine to putrescine, a rate-limiting step of the polyamine biosynthesis pathway [24]. Polyamines, mainly putrescine, spermine and spermidine play essential roles in transcription, translation, post-translational modification and cell signaling, and are important for cell growth and development [25]. It was previously reported that putrescine, the product of Odc1, peaks early following PHx and plays an essential role in hepatic regeneration in rat (*Rattus rattus*) [26]. The marked up-regulation of *odc1* shortly after PHx suggests that putrescine may also play critical roles in liver regeneration in zebrafish. Another top gene up-regulated early, *ptp4a2b* encodes a protein tyrosine phosphatase and belongs to the phosphatase of regenerating liver (PRL) family [27]. The *Ptp4a1* gene of rats was previously revealed to be an immediate early gene in the nucleus of regenerating liver [28]. The up-regulation of these regeneration-associated genes indicates similar regulatory mechanisms between mammalians and zebrafish. However, the *cog2* gene, which encodes a subunit of the conserved oligomeric Golgi complex and is involved in Golgi organization and intra-Golgi vesicle-mediated transport [29], has not been previously reported to be associated with regeneration.

GO enrichments for the DEG clusters with different expression profiles shed new light on the biological regulations underlying liver regeneration. The overrepresentation of the GO biological process term hormone activity by the genes transiently up-regulated from 6 h to 1 d (cluster 1) suggests important roles for hormones in regulating zebrafish liver regeneration after the PHx surgery. Activities of the endocrinal hormones including norepinephrine, growth hormone, insulin and thyroid hormones are required for liver regeneration [30]. These hormones can strongly induce and promote hepatocyte proliferation by triggering the associated signaling pathways, cytokines, growth factors and TFs [30]. In total, eleven genes in cluster 1 are associated with hormone activity (Appendix A). The hormones (encoding genes) include adiponectin (*adipoqb*), adrenomedullin (*adma*), apelin (*apln*), calcitonin (*calca*), erythropoietin (*epoa*), glucagon (*gcga*), hepcidin (*hamp*), leptin (*lepb*), natriuretic peptide (*nppb, nppc*) and relaxin (*rln1, rln3a*). Among the associated hormones, adiponectin, adrenomedullin and leptin were previously reported to promote liver regeneration in the mammalian PHx models [31,32,33,34,35]. However, negative effects in liver regeneration were documented for *apelin* and *hepcidin* [36,37]. The blocking activity of the apelin-APJ system could promote liver regeneration by activating Kupffer cells in the mouse (*Mus musculus*) PHx model [36]. The expression of hepcidin was inhibited by hepatocyte growth factor in the late stage of liver regeneration and overexpressing hepcidin-1 in mice impaired hepatic regeneration after PHx [37]. The transient up-regulation of the genes encoding hormones (*apelin* and *hepcidin*) with negative effects on liver regeneration in mammals represent stage- and species-specific responses to the liver injury in zebrafish. Except for the hormone-encoding genes mentioned above, the others that have not yet been linked with liver regeneration represent novel targets for further investigation.

Liver regeneration is a complex and tightly-regulated process [35]. One prominent feature of change in gene expression during zebrafish liver regeneration is that most DEGs were transiently up- or down-regulated. In addition to the genes up-regulated in the early stages after PHx, there are also late responsive genes (cluster 3). The genes of cluster 3 were enriched in glutathione metabolism, antioxidant activity, cellular detoxification and fatty acid oxidation. All of these are important physiological functions of the liver. It seems that the up-regulation of these genes represents a compensation for the impaired liver functions or functionalization of the regenerated hepatic tissues. In fact, PHx and subsequent liver regeneration can lead to oxidative stress [38]. Liver regeneration can be impaired by permanent oxidative stress and the activation of the cellular antioxidant responses has been shown to improve the process of liver regeneration [39]. Moreover, glutathione depletion depresses liver regeneration in the rats [40]. One of the most remarkable physiological alterations caused by PHx in the liver is the temporary accumulation of hepatic lipid, also called transient regeneration-associated steatosis [41]. This process is also essential for liver regeneration by providing energy through fatty acid β-oxidation [41]. Therefore, the biological processes enriched among the genes up-regulated at later stages are also required for liver regeneration.

The results of KEGG pathway enrichment analysis revealed the pathways involved in zebrafish liver regeneration. The most remarkable pathway enrichment for the up-regulated genes was the proteasome, which was overrepresented by the up-regulated genes throughout the experiment. The proteasome plays pivotal roles in many biological processes, such as transcription, signaling, protein trafficking and quality control and the cell cycle, through degrading cellular proteins in a tightly controlled manner [42]. PSMD2, a component of the proteasome (a non-ATPase subunit of the 19S regulator lid), was reported to promote the proliferation of HepG2 cells by facilitating lipid droplet accumulation in the cells [43]. The results of Western blotting confirmed an increase in the Psmd2 protein level during liver regeneration. These data prompt us to hypothesize that the activity of the proteasome is needed for liver regeneration. To test the hypothesis, we applied the proteasome inhibitor MG132 to the fish after PHx via intraperitoneal injection and analyzed the effect on the abundance of Pcna (used as a marker for hepatocyte proliferation). Unexpectedly, the injection of MG132 enhanced the Pcna level in the regenerating liver. MG132 has been reported to play a protective role against liver injury induced by ischemia/reperfusion in rats [44]. The mechanisms underlying the effect of MG132 in promoting liver regeneration remain to be further investigated.

## 4. Materials and Methods

### 4.1. Fish Lines

Wild type zebrafish of the AB line and the *Tg(fabp10a:dsRed; ela3l:EGFP)* transgenic fish line were used in this study. The transgenic fish line was obtained from China Zebrafish Resource Center (Wuhan, China). Its hepatocytes are labeled with DsRed, making them suitable for liver development and liver regeneration studies. The fish were maintained in an aquatic system supplied with recirculating water as previously described [45]. Females of about 8 months old (average body weight 1.8 ± 0.3 g) were subjected to hepatectomy.

### 4.2. Partial Hepatectomy (PHx) and Sampling

Previous studies have shown that the transcriptomes displayed sexual dimorphism in the zebrafish liver [46] and that the circadian clock gated regulatory steps in DNA synthesis and mitosis [47,48]. To eliminate the influences by gender and the circadian clock, regenerating livers were derived from female zebrafish at Beijing time 5:00–6:00 pm. The surgery procedure was performed as described previously [17] and illustrated in Appendix A. For the experimental groups, the ventral liver lobe was carefully exposed and a 2.5 ± 0.5 mm piece was resected from the tip (1/3) of the ventral lobe, leading to local injury. Sham-operated animals were subjected to the same procedure excluding liver resection. All of the sham fish survived after the surgery, while the PHx group demonstrated a survival rate of about 95% (Appendix A), indicating the success of the surgery. At 6 h, 1 d, 3 d, 5 d and 7 d after surgery, the remaining ventral liver lobes (from the resection margin to the distal tip) were collected. Due to the small amount of tissue collected from each fish, tissue from 3–5 individuals was pooled and regarded as a sample. Three biological replicates were included for each time point. The sham samples were collected at 24 h after sham surgery. The animal protocol for this study was approved by the Institutional Animal Care and Use Committee of the Institute of Hydrobiology (Approval ID: E01F050101).

### 4.3. Library Construction and High-Throughput Sequencing

Total RNA was extracted using TRIZOL reagents from Invitrogen following the manufacturer’s recommendations. RNA concentration was measured using NanoDrop 8000 from Thermo Scientific. The quality of the RNA samples was assessed by agarose gel electrophoresis. The integrity of the RNA samples was confirmed using Agilent 2100 Bioanalyzer and 4 μg of total RNA was used for the isolation of mRNA. RNA sequencing library construction and high-throughput sequencing were performed by experts in the Analytical & Testing Center at the Institute of Hydrobiology, Chinese Academy of Sciences (http://www.ihb.ac.cn/fxcszx/, accessed on 28 June 2024) as described previously [49,50]. Multiplexed libraries were sequenced for 72 bp at both ends (paired-end) using an Illumina Genome Analyzer IIx platform according to the standard Illumina protocols as reported previously [51]. The sequencing data have been deposited in the NCBI Sequence Read Archive (SRA, http://www.ncbi.nlm.nih.gov/Traces/sra, accessed on 28 June 2024) under the accession number SRP053641.

### 4.4. Bioinformatic Analysis of RNA-seq Data

Raw reads were first processed by the prinseq-lite (v0.20.4) [52] to trim the low-quality bases and filter low-quality reads with the following parameters “-trim_qual_left 20 -trim_qual_right 20 -min_len 30 -min_qual_score 10 -min_qual_mean 20 -ns_max_n 0”. The clean reads were mapped to the reference transcriptome of zebrafish (GRCz11) using salmon (v1.10.0) [53] with default parameters. Abundances for the genes (summarization of transcript abundances) were calculated by tximport (1.30.0) [54] and expressed as transcript per million (TPM). The genes with a TPM value ≥ 1 in all of the samples of at least one experimental group were regarded as expressed. The read count table was submitted to DESeq2 (v1.42.0) [55] for gene differential expression analysis and genes differentially expressed between the regenerating liver samples at different time points after resection and the sham controls were identified. The thresholds for DEG identification were fold change > 1.5 and adjusted *p* value < 0.05. Principal component analysis for the gene abundance data was conducted using ArrayTrack [56]. The differentially expressed genes were clustered using K-means analysis of cluster 3.0. Clustering results were visualized using JavaTreeview (http://jtreeview.sourceforge.net/manual.html, accessed on 1 June 2024) [57].

Weighted gene co-expression network analysis for the genes in the representative clusters was performed using WGCNA (v1.72.5) with parameters “power 12, minModuleSize 30, mergeCutHeight 0.2” [58]. The major module with the largest number of genes for each cluster were subjected to subsequent analysis.

Gene ontology (GO) and the Kyoto Encyclopedia of Genes and Genomes (KEGG) pathway enrichment analyses were conducted using Cytoscape (v.3.9.1) plugins BiNGO (v.3.0.5) [59] and ClueGO (v2.5.10) [60], respectively. The ontology and annotation files for GO enrichment analysis were downloaded from the gene ontology website (http://www.geneontology.org/, accessed on 20 May 2022) and the database used for KEGG pathway enrichment analysis was released on 25 May 2022. For the enrichment analyses, all of the expressed genes were used as reference.

### 4.5. Real-Time Quantitative PCR

Real-time quantitative PCR (qPCR) was performed as described previously [61] to validate the results of RNA-seq. Briefly, total RNA samples were treated with RNase-free DNase I (Promega) to eliminate contaminated genomic DNA before reverse transcription. First-strand cDNA for each sample was synthesized from 1 μg of treated total RNA using a random hexamer primer within the RevertAidTM First Strand cDNA Synthesis Kit from Fermentas. The qPCR primers were designed using Primer3Plus (https://www.primer3plus.com/, accessed on 28 June 2024). Real-time quantitative PCR was conducted using a CFX ConnectTM Real-Time PCR detection System from BioRad. The amplification was carried out in a reaction system of 20 μL including 10 μL of 2 × SYBER Green Real Time PCR Master mix from BioRad, 2 pmol of each primer and 5 μL of 10 × diluted cDNA template. Three biological replicates of the sham and regenerating livers at different time points were included in the analysis. All reactions were carried out in duplicate. The amplification program was 95 °C for 1 min, followed by 39 cycles of 95 °C for 10 s, 55–60 °C for 30 s (with plate read) and 72 °C for 10 s. The melt curve of the PCR product was generated by denaturization at 95 °C for 10 s, heating from 65 °C to 95 °C with 0.5 °C increments and a 5 s dwell time, and a plate read at each temperature. The specificity of the reaction product was confirmed by the observation of a single peak for the melt curve. The amplification cycle displaying the first significant increase in the fluorescence signal was defined as the threshold cycle and used for quantification (Cq).

Before qPCR analysis, the standard curve for the primers was generated by analyzing a series of 5-fold diluted standard samples prepared by combining equal amounts of all of the samples to be analyzed. The amplification efficiency of primers was calculated from the slope of the corresponding standard curve. Information including the accession number of the nucleotide sequence, primer sequences and amplicon length were listed in Appendix A. The *b2m* gene was used as an internal reference for the normalization of gene expression. The mean normalized expression of target genes was calculated using the Q-gene software [62].

### 4.6. MG132 Treatment

After the PHx surgery, each individual was intraperitoneally injected with 10 µL of 1 mg/L MG132 solution (#S2619, Selleck). The controls were injected with 10 µL of 0.1% DMSO solution. Samples were collected at 3, 5 and 7 d after the operation as described above.

### 4.7. Western Blotting

Western blotting was performed as previously described [63] to characterize the protein levels of Psmd2 and Pcna during the regeneration of the intersected liver. Levels of the target proteins were normalized to that of β-actin. Briefly, liver samples were homogenized in the lysis buffer supplemented with 1% protease inhibitor cocktail (#B14001, Bimake) and 1 mM PSMF (#ST507, Beyotime). The 10% PAGE gel kit (#PG112) from Epizyme was used to prepare a gel for the electrophoresis of the samples. The proteins in the gel were transferred to a 0.45 µm PVDF membrane (#IPVH00010, Merk Millipore). Antibodies used for the immunodetection of the target proteins included anti-PSMD2 (#GB113525-100, Servicebio), anti-PCNA (#GB11010-100, Servicebio) and anti-β-actin (#AC026, ABclonal). Finally, the membranes were incubated with the chemiluminescent substrate (#p90719, Merk Millipore) and images were taken using a chemiluminescence imaging system (#SCG-W3000) from Servicebio.

### 4.8. Statistics

SPSS 15.0 software for windows was used for statistical analysis. The data for gene expression were analyzed by the independent-samples t-test. The correlation between the data of RNA-seq and qPCR was analyzed by the Spearman’s rho test. The data were expressed as mean ± standard deviation (SD).

## 5. Conclusions

Gene expression during liver regeneration after PHx was explored by RNA-seq. In total, 12,809 genes were found to be expressed in the liver samples. Most of the genes affected by the PHx surgery demonstrated transient up- or down-regulation at certain stages after the injury. The GO biological processes and KEGG pathways associated with liver regeneration were identified. Inhibiting proteasome activity by the application of MG132 to the fish promoted liver regeneration after PHx. Our data shed new light on the molecular mechanisms underlying liver regeneration.

## Figures and Tables

**Figure 1 ijms-25-07212-f001:**
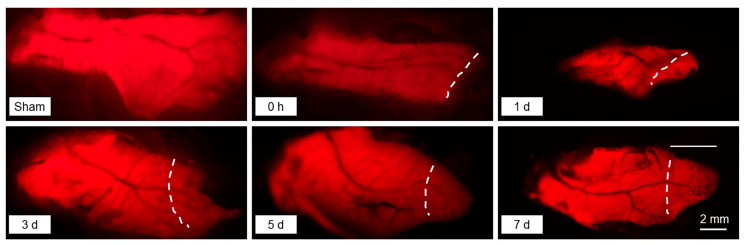
Representative photos for the regenerating liver at the indicated times after the PHx surgery. The white dashed curves indicate the cutting edge resulting from the PHx surgery. The regions to the right of the curves are regenerated tissue after hepatectomy. The *Tg(fabp10a:dsRed; ela3l:EGFP)* transgenic fish line specifically expressing DsRed in the liver was used for the experiment.

**Figure 2 ijms-25-07212-f002:**
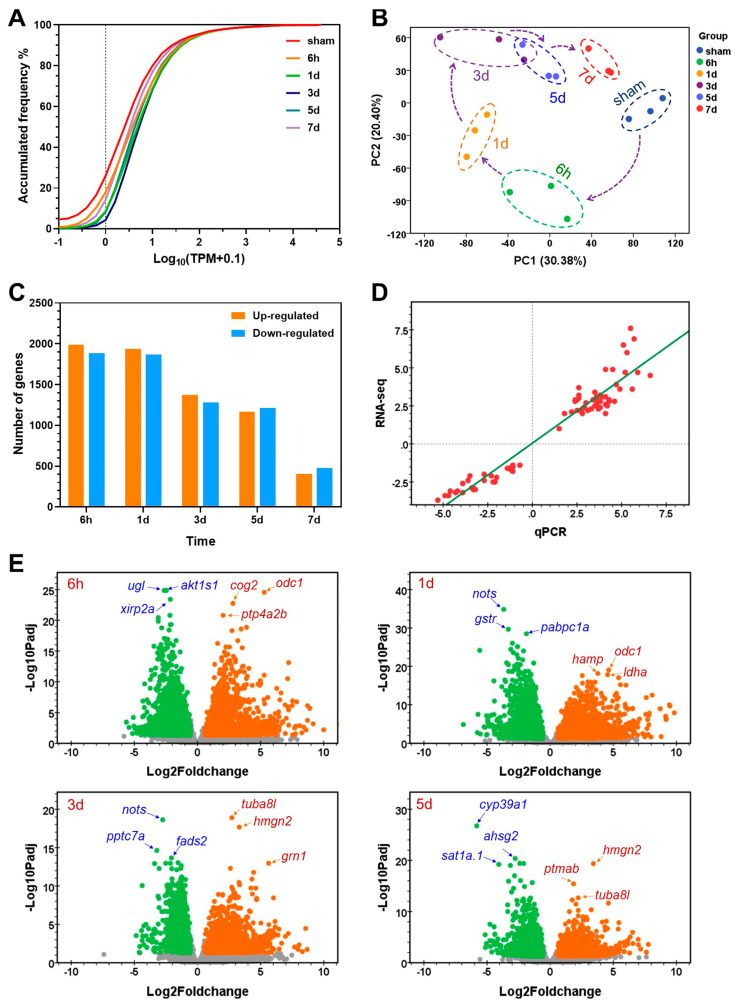
Overall changes in gene expression after partial ventral lobe hepatectomy in zebrafish. (**A**) Frequency distribution of gene transcriptional abundance for the experimental groups. The dashed line indicates a cumulative ratio of the low abundance genes under the indicated conditions. (**B**) Principal component analysis of the gene abundance data. (**C**) Numbers of the up- and down-regulated genes at the indicated time points after the PHx surgery. (**D**) Correlation between the qPCR and RNA-seq data. (**E**) Volcano plots demonstrating the DEGs at different time points. Yellow, up-regulated; green, down-regulated; gray, unchanged. Names of the most significantly up- (red) and down-regulated (blue) genes are shown.

**Figure 3 ijms-25-07212-f003:**
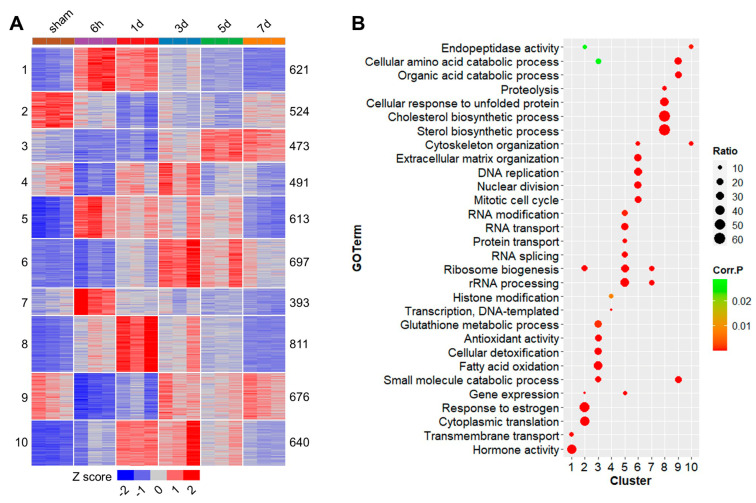
Cluster analysis for the DEGs and GO enrichments, network eigengene values and hubs for the gene clusters. (**A**) A heatmap demonstrating results of K-means clustering for the DEGs. The cluster numbers (left) and numbers of DEGs contained in the clusters (right) are also shown. (**B**) Gene ontology enrichments for the DEG clusters. Ratio, ratio of the identified genes relative to all of the genes in the genome associated with the GO term; Corr. P, corrected *p* value for the enrichment analysis.

**Figure 4 ijms-25-07212-f004:**
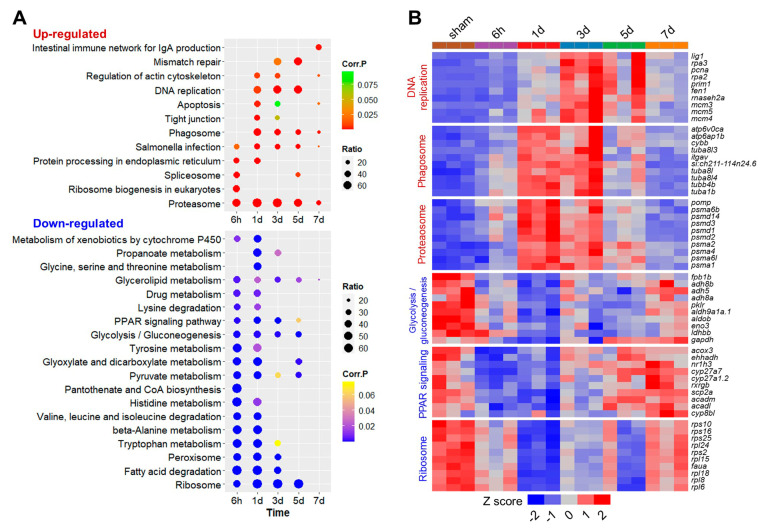
KEGG enrichments for the up- and down-regulated genes after the PHx surgery. (**A**) Bubble plots illustrating the KEGG pathway enrichments for the DEGs. (**B**) Heatmaps illustrating expression profiles of the representative genes associated with the top pathways enriched for the up- (red) and down-regulated (blue) genes. Pathway names are shown at the left and gene names are shown at the right.

**Figure 5 ijms-25-07212-f005:**
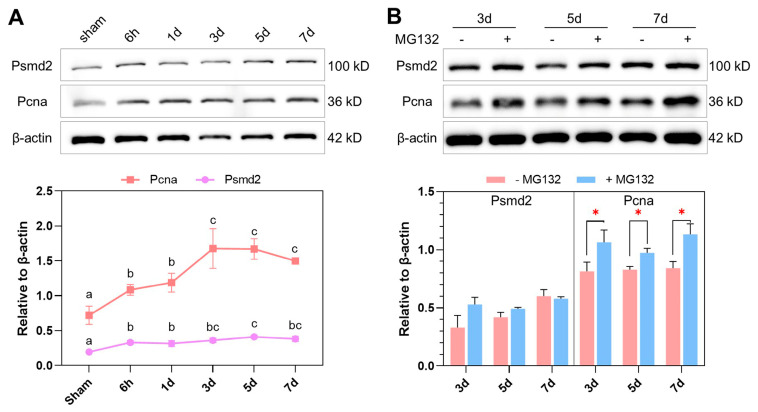
Effects of modulating proteasome activity on liver regeneration. (**A**) Representative Western blots demonstrating levels of Psmd2 and Pcna proteins during liver regeneration after the PH surgery. (**B**) Representative Western blots demonstrating effects of MG132 on Psmd2 and Pcna expression during liver regeneration after the PHx surgery. The line chart and bar chart below the Western blots demonstrate the abundance of the target proteins relative to that of β-actin. Different letters above the error bars denote significant differences among the means (*p* < 0.05, n = 3, one-way ANOVA followed by Duncan’s test); *, *p* < 0.05, n = 3 (independent samples *t*-test).

## Data Availability

The data presented in this study are available in the article and Appendix A.

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
