# Peer review of "Transcriptomic Characterization of Key Factors and Signaling Pathways for the Regeneration of Partially Hepatectomized Liver in Zebrafish"

_ijms, 2024, doi:10.3390/ijms25137212_

Round 1
Reviewer 1 Report
Comments and Suggestions for Authors
The paper is very interesting. Altough there are some minor issues that should be improved:
Introduction
Lines 93-102 This is the aim of the study but it sounds like a part of methodology chapter. It should be rephrased.
Results
Lines 105-108 This should be transferred to “Method” chapter. This is a description of preparing samples and efficiency of this process. Also Fig A and B should be transferred (the order of Figs will be changed, so their numbers/letters order should be changed)
Lines 127-128 First sentence should be removed. Authors should avoid methodology sentences in the “Results” chapter.
Overall – the “Results” chapter could be more concise. Authors should avoid repeating discussing all results from Tables/Figs. Only most relevant should be emphasised in the text and then discussed.
Conclusions
Line 613 –627 those sentences present once again the results not conclusions. It should be written once again. Similarly to conclusions made in “Abstract” chapter
Author Response
Please· see the attached file.

Reviewer 2 Report
Comments and Suggestions for Authors
Song et al have performed a timecourse analysis of regenerating zebrafish transcriptome following partial hepatecomy. The study is scientifically acceptable, although improvements could be made (see minor points, below, concerning sample pooling, n, and statistics) and the data has been loaded onto a public repository for release upon publication, so this is a potentially valuable contribution to the field. Therefore, the work should be published. However, improvements are needed, as detailed below.
Major problems
1) Incorrect statements. The authors claim, both in the abstract and the introduction, that partial hepatectomy is “…one of the most efficient methods for treating liver diseases…”. This is a ridiculously wrong statement and, in the introduction where references can be given, none are. The preceding statement in the introduction r.e. cirrhosis is also innacurate and not supported by citations. Unacceptable and unnecessary. Statement about hepatectomy leading to “liver mass loss of 30%” is also wrong. This should be loss of 70%.
2) This is an extremely inefficient presentation. Experimentally, the authors performed only a transcriptome time course and a single drug (MG132)-challenge followed by a western blot for 2 genes. The data from this are good-enough and have valuse. However, the authors followed this with mountains of bioinformatics analyses of the transcriptomes to uncover dozens of potentially intriguing patterns in the data, none of which are tested or validated. This has resulted in 21 pages of highly speculative and untested correlations. This has little value and encumbers the paper, lowering its value. I highly recommend that the vast majority of informatic analyses that do not lead to compelling mechanistic insights, and the discussions concerning these, be moved to Supplementary materials.
3) The study appears to have not established a reasonable difference-threshold for defining noteworthy up- or down-regulated genes, and magnitude differences appear to be considered only in Fig. 2E (…perhaps 2D, but there are no units on the axis, so this is uninterpretable). Importantly, the sum 24 genes mentioned in the transcriptome timepoints in Fig. 2E do not include any of the transcripts that are “hallmarked’ in the abstract and text (cirbpb, hif1ab, klf6, uhrf1, top2a, capin3b, other). What is going on here? What magnitude differences are included in analyses - 2-fold cutoff or 1.2-fold? It seems the authors are contriving pathways, stories, and hypotheses on potentially very minor gene expression differences. This needs to be much more clearly
Specific major points.
1. The text says the zebrafish hepatectomies were 1/3; however the descriptions in the M&M section say they are ~9%. Please clarify.
2. The methods state that, although only 4 µg of RNA was used for each RNAseq reaction, livers from 3 fish were pooled for each dataset, and 3 datasets were collected for each condition. This is a disappointing design. Pooling masks the important variation and n=3 replicates for each condition is horribly low. The power of this data would be much much higher if the authors would have assessed individual animals, not pools, and thereby had n=9 for each condition. This was not well thought-out.
3. As presented, and perhaps intrinsically, Fig 3D has no value. What do the connecting lines signify? Are these published protein-protein interactions, metabolic pathway connections, etc? Please define “betweenness centrality”, explain how the colors are proportional to this quality, and provide a scale.
4. The authors say they used wb analyses of pcna and psmd2 to assess proteasome activity. psmd2 is not defined. pcna is a replication complex factor and not, to my knowledge, a measure of proteasome activity.
5. MG132 treatment resulted in increased pcna expression. The authors interpret this as MG132 “promoting liver regeneration”. I interpret this as MG132 increasing liver cell death. Please provide data that discriminates between these 2 possibilities.
6. The section on alternative splicing highlights cirbpb, which shows minor alternative splicing at the 3’ end. The discussion includes no information about what cirbpb is, what is known about its activities, or how this alternative splicing might be expected to alter the domain-structure or activity of the protein. There is no empirical data to support any role for cirbpb in regeneration. Like most of the genes hallmarked in this manuscript, the authors show correlations with no empirical testing, no validation, no measure of the levels of expression or magnitudes of differential expression (or differential splicing), and follow this with extensive speculative conjecture about these genes potentially playing important roles in liver regeneration.
7. Fig 5 (western blots) have n=1 for each condition and no significance can be ascribed to any of the data. This data, and all discussion of it (e.g., discussion of MG132 or proteasome) therefore have no value. The experiment sufficient replicates to allow statistical evaluation.
Specific minor points.
1. All gene or protein acronyms should be defined at first use, and a description of what is known about each should be provided. What, e.g., are “cirbpb, hif1ab, and klf6”, mentioned in the abstract?
2. No units on chart in Fig. 2D.
3. Define the transgenic line, including the promoters for both fluorescent reporters. Where is GFP expressed and why is this ignored in the paper.
4. Why does regeneration cause the liver to become brighter red? Why is this significant in the context of this paper (i.e., is Fig. 1C valuable?) Since the transgenic fish express these fluorescent reporters, these should, it seems, provide internal controls for RNAseq data. I did not see this data presented or used for validation.
5. Fig. 1B has little or no value. At best this should be supplementary. Fig. 1A could also be supplementary.
6. The authors state that expression of 12,809 “genes” was detected in the study, but “genes” is not defined. Does this include all splice isoforms? By the definition used, how many “genes” are in the z.f. genome?
Summary: The transcriptome data, although disappointingly only n=3 of pooled animals, and the descriptive analyses of these data, have value. In combination with release of these data through NCBI SRA, this supports publication of this work. However, the exceptionally long and speculative presentation, and inclusion of large numbers of low-value or inadequately described figure panels, is not acceptable. A revised manuscript should be prepared that moves most of these items to supplementary and ensures they are clearly described. Also, for any specific genes or pathways that the authors choose to hallmark in the abstract or text, I will expect to see independent validation of each of these components being meaningful contributors to the underlying biology of liver regeneration in zebrafish presented.
Comments on the Quality of English Language
No concerns
Round 2
Reviewer 2 Report
Comments and Suggestions for Authors
The review comments have been adequately addressed and the revised manuscript is considerably improved. I recommend acceptance for publication.